# Surgical Site Infection after Bone Tumor Surgery: Risk Factors and New Preventive Techniques

**DOI:** 10.3390/cancers14184527

**Published:** 2022-09-19

**Authors:** Shinji Miwa, Norio Yamamoto, Katsuhiro Hayashi, Akihiko Takeuchi, Kentaro Igarashi, Hiroyuki Tsuchiya

**Affiliations:** Department of Orthopedic Surgery, Graduate School of Medical Science, Kanazawa University, Kanazawa 920-8640, Japan

**Keywords:** bone tumor, surgical site infection, risk factors

## Abstract

**Simple Summary:**

Surgical site infection (SSI) is a serious complication of the surgical treatment of malignant bone tumors. Malignant bone tumor surgeries have higher rates of SSI than other orthopedic surgeries. In patients with SSIs, additional surgeries, long-term administrations of antibiotics, extended hospital stays, and the postponement of scheduled adjuvant treatments are required. Therefore, SSIs may deteriorate functional and oncological outcomes. To improve the surgical outcomes of patients with malignant bone tumors, preoperative risk assessments for SSIs, new preventive techniques against SSIs, and the optimal use of prophylactic antibiotics are required. Recently, various studies have shown significant associations between SSIs and age, tumor site (pelvis and tibia), extended operative time, the use of implants, body mass index, leukocytopenia, and reconstruction procedures. Furthermore, prophylactic techniques, including silver and iodine coatings on implants, have been developed. In this review, the predictive factors of SSIs and new prophylactic techniques are discussed.

**Abstract:**

The management of malignant bone tumors requires multidisciplinary interventions including chemotherapy, radiation therapy, and surgical tumor resection and reconstruction. Surgical site infection (SSI) is a serious complication in the treatment of malignant bone tumors. Compared to other orthopedic surgeries, the surgical treatment of malignant bone tumors is associated with higher rates of SSIs. In patients with SSIs, additional surgeries, long-term administrations of antibiotics, extended hospital stays, and the postponement of scheduled adjuvant treatments are required. Therefore, SSI may adversely affect functional and oncological outcomes. To improve surgical outcomes in patients with malignant bone tumors, preoperative risk assessments for SSIs, new preventive techniques against SSIs, and the optimal use of prophylactic antibiotics are often required. Previous reports have demonstrated that age, tumor site (pelvis and tibia), extended operative time, implant use, body mass index, leukocytopenia, and reconstruction procedures are associated with an increased risk for SSIs. Furthermore, prophylactic techniques, such as silver and iodine coatings on implants, have been developed and proven to be efficacious and safe in clinical studies. In this review, predictive factors of SSIs and new prophylactic techniques are discussed.

## 1. Introduction

The treatment of malignant bone tumors requires multidisciplinary interventions such as surgical tumor resection, chemotherapy, and radiation therapy. After surgical tumor resection, most bone defects require reconstruction using endoprostheses, allografts, autografts, artificial bones, or distraction osteogenesis [1,2,3,4,5]. During the surgical treatment of malignant bone tumors, risk conditions such as soft-tissue defects, large implants, allografts, frozen or irradiated bone, and radiation-therapy- or chemotherapy-induced immunodeficiency may cause complications (including surgical site infection (SSI), the loosening and breakage of the implant, delayed union, and the fracture of the grafted bone). SSI is one of the most severe complications related to bone tumor surgeries. The incidence of SSI following malignant bone tumor surgery has been reported to be 3–10 times higher than that following general orthopedic surgery [6,7]. The incidence of SSI in patients who underwent orthopedic surgeries was reported as 0.2–4.2% [8,9,10,11,12,13]. The incidence of SSI in total hip arthroplasty was reported as 0.2% [13], whereas the incidences of SSI in spine surgeries were reported as 3.4–4.2% [10,12]. On the other hand, high incidences of SSI have been reported in patients with malignant bone tumors [14,15,16,17,18,19,20,21,22,23,24,25,26,27,28,29]. In patients who underwent megaprosthesis after malignant bone tumor resection, 9.3–20.4% of the patients had SSI [14,15]. SSI rates after biological reconstruction were reported as 11.9–13.5% [16,17]. In patients with SSI, additional treatments (such as the long-term administration of antibiotics, implant and grafted bone removal, debridement, irrigation, and reconstruction) are often required. Therefore, SSIs entail longer periods of hospitalization, functional deteriorations, excessive medical costs, and delays in the treatment course, which increase mortality [30,31,32]. 

During bone cancer surgery, surgical reconstruction of the bone, ligaments, tendons, nerves, and vessels requires a long operative time and biomaterials, which may increase the risk of SSIs. Therefore, a preoperative assessment of the risk of complications is needed, and appropriate surgical procedures and adjuvant treatments should be chosen based on the risks and benefits of the surgery. In addition to the rarity of malignant bone tumors, the diversity of tumor locations, surgical procedures, biomaterials, and neoadjuvant or adjuvant treatments makes it difficult to identify risk factors for SSI. However, several studies have demonstrated predictive factors for SSI in patients with bone tumors. Furthermore, new techniques of antibacterial coating on implants have been developed, and their usefulness has been reported. This review article summarizes the predictive factors of SSI and recent studies on the techniques for antibacterial coating on implants.

## 2. Diagnosis of SSI

There are various definitions of SSI in clinical practice. Bruce et al. reported that there are numerous definitions of SSI, five of which were defined by expert groups, including the Centers for Disease Control (CDC), the Public Health Laboratory Service, the Surgical Infection Society Study Group (SISG), and the second National Prevalence Survey study group (NPS) (Table 1) [33,34,35,36,37,38,39]. Among these definitions, the CDC criteria are the most widely used to diagnose SSI [38]. According to the CDC criteria, SSIs are classified into superficial and deep infections. A superficial SSI is defined as an infection that involves only the skin or subcutaneous tissue of the incision, occurs within 30 days postoperatively, and has at least one of the following features: purulent drainage from the superficial incision, the isolation of organisms from a culture specimen taken from the superficial incision, at least one symptom/sign of infection (pain/tenderness, swelling, redness, and local heat), superficial incision intentionally opened by the surgeon, or a diagnosis of infection by a physician or surgeon. A deep SSI is defined as an infection that involves the deep soft tissues of the incision, occurs within 30 days after surgery without the use of an implant or within 1 year after the surgery with the use of an implant, and has at least one of the following features: purulent drainage from the deep incision, spontaneous incisional wound dehiscence or deliberate opening of the incision by the surgeon in the presence of pain/fever/tenderness, an abscess or other evidence of infection involving the deep incision, and the diagnosis of deep SSI by a physician or surgeon. An organ/space SSI is defined as an infection that involves organs or spaces and occurs within 30 days after a surgery without the use of an implant or within 1 year after a surgery involving the use of an implant. It should have at least one of the following features: purulent drainage from a drain that is placed into the organ/space, the isolation of organisms from a culture from the organ/space, an abscess or other evidence of infection involving the organ/space, or the diagnosis of an organ/space SSI by a physician or surgeon.

## 3. Risk Factors for SSI

### 3.1. Age and Gender

In a study on the risk factors for SSI in 1304 patients who underwent musculoskeletal tumor excision, age was identified as an independent risk factor for SSI (OR = 1.18; *p* < 0.001, Table 2) [41]. In another study involving 45 patients who underwent pelvic tumor resection, age also was significantly associated with SSI (OR = 1.04, *p* = 0.032) [18]. In contrast, there was no significant difference in age between patients with and those without SSIs in a different study involving 757 patients with bone and soft-tissue tumors [42]. Furthermore, several other studies on the risk factors for SSI in bone tumor surgeries showed no significant association between age and SSI [26,43,44]. In studies on SSI in particular sites, age had no significant association with SSI in patients with bone tumors in the pelvis, lower limb, and knee [27,45,46]. Based on these studies, age does not seem to be a strong predictor for SSI in bone tumor surgeries.

A few reports have suggested an association between sex and risk for SSI in bone tumor surgeries. Aponte-Tinao et al. investigated the risk factors for deep infections in 673 patients who underwent reconstruction with a massive allograft in the long bones [47]. In this study, 60 (9%) patients developed an infection, and male sex (OR = 1.92; *p* < 0.029), was associated with an increased risk of infection. Contrarily, other studies have shown no significant association between sex and the risk of SSI [42].

### 3.2. Diabetes and Nutritional Condition

Richards et al. investigated the correlation between hyperglycemia and 30-day SSI in 790 patients with orthopedic trauma [48]. In this study, multivariate logistic regression models revealed that a blood glucose levels of ≥ 200 mg/dL was a risk factor for SSI (odds ratio (OR) = 2.7) after adjusting for open fractures (OR = 3.2). In a meta-analysis of the risk factors for SSI after spinal surgery, diabetes (risk ratio = 2.22, *p* = 0.001) and body mass index (BMI) > 35 (risk ratio = 2.36; *p* < 0.001) were significantly associated with an increased risk of SSI [49].

Some studies on musculoskeletal tumors have also shown associations between the nutritional condition, BMI, and the risk of SSI [42]. In a prospective study involving 110 patients who underwent surgery for lower extremity or pelvic tumors, obesity was an independent predictor of SSI in a multivariate analysis (*p* = 0.016); however, no significant association was observed between diabetes and SSI [50]. Anatone et al. reported that 14% of patients with SSI and 8% of those without SSI had diabetes (*p* = 0.051) [42]. Lozano-Calderón et al. reported that BMI (OR = 1.40, *p* < 0.05) was significantly associated with deep infection in 32 patients who underwent reconstruction with allograft after proximal tibia resection [51]. Based on these studies, BMI and nutritional status can be considered to be risk factors for SSI in bone tumor surgery.

### 3.3. Tumor Location

Tumor location is considered a predictor of SSI in cancer surgeries. In a prospective study involving 504 elective cancer surgeries, the groin location (OR = 4.7) and head/neck locations (OR = 0.1) were significant predictors of SSI [52].

Because a high incidence of SSI (10–37%) has been reported in pelvic tumors, this location is thought to be associated with an increased risk of SSI [18,19,20,21,22,23,24,25,26,27,28,29]. In a study involving 1304 patients with musculoskeletal tumors, the hip region was reported as an independent risk factor for SSI (OR 1.96, *p* < 0.001) [41]. In another study involving 757 patients with bone and soft-tissue tumors, in 16% of patients in the non-SSI group and 26% of patients in the SSI group, the tumor location was the pelvis/hip region (*p* = 0.070) [42]. In a retrospective study on bone tumors, pelvic tumors were associated with an increased risk of SSI (OR = 3.42; *p* = 0.044) [43]. In a study on surgical complications in 17 patients treated with saddle prosthesis, 18% had deep infections, 23% had migration of the saddle, 18% had saddle dislocations, and 12% had sacroiliac subluxations [22]. In a study involving 98 patients who underwent tumor resection and reconstruction with pelvic endoprosthesis for periacetabular tumors, high complication rates were observed; these included local recurrences (31%), infections (30%), and dislocations (20%) [20]. In another study involving 35 patients with periacetabular malignant tumors who underwent tumor resection and prosthetic reconstruction, there was a high incidence of complications, including deep infection (26%), local recurrence (24%), and hip dislocation (17%) [19]. Karaca et al. investigated the incidence of SSI in 68 patients who underwent hemipelvectomy [53]. In their study, 34 (50%) patients had SSI, and the infection rates of external hemipelvectomy and internal hemipelvectomy were 52% and 17%, respectively (*p* = 0.02).

The incidence of SSI in patients with tibial tumors has been reported to be 10–32% [44,54,55]. Lozano-Calderón et al. reported a high incidence of soft-tissue complications (48%) in patients who underwent wide tumor resection and reconstruction of the proximal tibia; they found that four of five deep infections occurred in patients with soft-tissue complications [51]. In a study on massive allograft reconstruction in long bones, tibial tumors were associated with an increased risk of deep infections (OR = 3.17; *p* < 0.001) [47]. Furthermore, Langit et al. investigated the risk factors for SSI in 256 patients with malignant bone tumors of the extremities and reported that tumor localization in the tibia (OR = 6.04; *p* < 0.001) was an independent risk factor for postoperative deep infection [44].

### 3.4. Biomaterial

In general surgery, the use of biomaterials is thought to be a risk factor for SSI [32]. Furthermore, high infection rates of 9–28% have been reported in patients after reconstruction using endoprosthesis, against low rates of 0.9–1.2% in patients in whom implants were not used [56,57,58,59,60,61]. In a study involving 681 patients with bone tumors who underwent tumor excision, a multivariate analysis showed that implant use was an independent risk factor for SSI (OR = 9.28; *p* = 0.006) [43]. In another study involving 1304 patients with musculoskeletal tumors, the existence of an implant/bone graft (OR = 1.94; *p* = 0.001) was shown to be an independent risk factor for SSI [41]. In a study on pelvic tumors, 26% of patients treated with reconstruction had SSI, whereas 15% of patients treated without reconstruction had SSI (OR = 0.51, *p* = 0.033) [27]. Based on these studies, the use of biomaterials can be considered to be a risk factor for SSI in bone tumor surgery.

### 3.5. Intraoperative Blood Loss

Although intraoperative blood loss is also thought to be a risk factor for SSI in orthopedic surgery, no report has shown the significance of intraoperative blood loss on SSI in bone tumor surgery. Anatone et al. reported that the estimated blood loss volumes in patients with and without SSI were 1237 mL and 191 mL, respectively (*p* < 0.001) [42]. Severyns et al. reported that intraoperative blood transfusion was a risk factor for SSI (OR = 1.12, *p* = 0.019) [18]. In other studies, no significant association between intraoperative blood transfusion and SSI was observed in multivariate analyses [26,41]. In a prospective study involving 110 patients who underwent surgery for lower extremity or pelvic tumors, blood transfusion was an independent predictor of SSI in a multivariate analysis (*p* < 0.007) [50].

### 3.6. Operative Time

Several studies have revealed a significant association between operative time and SSI. In a prospective study involving 504 elective cancer surgeries, operative times ≥ 2 h (OR = 1.8) and ≥4 h (OR = 2.2) were significant predictors of SSI [52]. The National Nosocomial Infections Surveillance (NNIS) system has reported that an operative time ≥ 4 h is a risk factor for SSI in general surgery [62]. Furthermore, a meta-analysis on spinal surgery revealed a significant association between an operative time > 3 h (risk ratio = 2.16; *p* = 0.009) and SSI [49]. In a study on musculoskeletal tumor surgeries, the surgical times in patients with SSI and those without SSI were 107 min and 190 min, respectively (*p* < 0.001) [42]. In another study on musculoskeletal tumor surgery, the number of preceding procedures (OR = 1.2, *p* < 0.01) and the procedure duration (OR = 1.2, *p* < 0.01) were independent risk factors for SSI [41]. In a study involving 302 patients who underwent malignant bone tumor excision and reconstruction, an operative time ≥ 5 h (OR = 3.4, *p* = 0.022) was an independent risk factor for SSI [26]. Another study revealed that an operative time ≥ 5 h was an independent risk factor in 256 patients who underwent surgical excision for malignant bone tumors of the extremities (OR = 3.3, *p* = 0.027) [44]. Because a long operative time is a strong risk factor for SSI, shortening the operative time may be effective in reducing the incidence of SSI.

### 3.7. Radiation Therapy

Although tissue damage due to radiation therapy is thought to increase the risk of SSI, the association between radiation therapy and SSI remains controversial. Several studies have shown no significant association between radiation therapy and an increased risk of SSI [43,50]. Contrarily, Demura et al. reported that radiation therapy was a risk factor for SSI in patients with metastatic spinal tumors [15]. In a study on the risk factors for SSI in 1034 patients with musculoskeletal tumors, immediate preoperative radiation therapy (OR = 2.6, *p* < 0.001) and a remote history of radiation therapy (OR = 2.6, *p* < 0.0001) were significantly associated with an increased risk of SSI [41]. Although only a few reports have shown a significant association between radiation therapy and the risk of SSI, radiation therapy appears to be a strong risk factor for SSI in bone tumors. However, further studies on the impact of radiation therapy on SSI are required.

### 3.8. Chemotherapy

Although chemotherapy-induced neutropenia and leukocytopenia may increase the risk for SSI, the association between chemotherapy and SSI remains controversial. In a study on cancer surgeries, preoperative chemotherapy (OR = 1.9) was reported to be a significant predictor of SSI [52]. However, only a few reports have shown an association between chemotherapy and SSI in musculoskeletal tumor surgeries [42,50,51]. In a study on 1521 musculoskeletal tumor surgeries, neoadjuvant chemotherapy (OR = 1.8, *p* = 0.02) and postoperative chemotherapy within 1 month (OR = 2.2, *p* < 0.01) were significantly associated with SSI [41]. In another study investigating the characteristics of patients with SSI, 34% of patients in the SSI group and 13% in the non-SSI group had undergone neoadjuvant chemotherapy (*p* < 0.001) [42]. In a study on the predictive factors for deep infection in 32 patients who underwent reconstruction using an allograft after proximal tibia resection, a lower preoperative white blood cell count was identified as an independent predictor of deep infections [51]. In contrast, other studies have shown no significant association between chemotherapy and an increased risk of SSI [43,44,50]. Based on these studies, a decreased number of white blood cells can be considered to be associated with SSI.

### 3.9. Other Factors

Anatone et al. reported higher rates of smoking history (20% in patients without SSI vs. 30% in patients with SSI, *p* = 0.041), higher rates of malignant disease (32% in patients without SSI vs. 63% in patients with SSI, *p* < 0.001), and higher American Society of Anesthesiologists (ASA) scores (*p* = 0.002) in patients with SSI [42]. However, there was no significant association between the use of immunosuppressive drugs and the incidence of SSI. In a study involving 45 patients with pelvic tumors, the preoperative ASA score was found to be a risk factor for SSI (OR = 18.3, *p* = 0.001) [18]. Lee et al. reported that admission from a health-care facility (OR = 4.35) was a risk factor for SSI in elderly patients who underwent orthopedic surgeries [63]. In a study on reconstruction with massive allografts in long bones, procedures performed in a conventional operating room (OR = 3.15, *p* < 0.002) and the long-term use of postoperative antibiotics (OR = 2.25, *p* < 0.041) were associated with an increased risk of infection [47].

## 4. Current Practices in the Prophylaxis of SSI and Emerging Techniques

Although perioperative intravenous antibiotics are commonly administered to prevent SSI, there are still controversies regarding antibiotic regimens that can effectively prevent SSI. At the International Consensus Meeting on Musculoskeletal Infection, there was no recommendation to adjust the type, dose, and duration of antibiotic prophylaxis in patients undergoing oncologic endoprosthetic reconstruction from that which is routinely administered in conventional total joint arthroplasty [64]. A meta-analysis showed that the rate of SSI after endoprosthetic reconstruction in the lower extremity was 10%, and that the postoperative use of prophylactic antibiotics for a period longer than 24 h decreased the risk of SSI [65]. Contrarily, Ghert et al. conducted the Prophylactic Antibiotic Regimens in Tumor Surgery (PARITY) trial to compare the effects of a 5-day postoperative intravenous antibiotics regimen with those of a 1-day regimen on the prevention of SSI and antibiotics-related adverse effects in patients who underwent tumor resection and endoprosthetic reconstruction of the lower extremity [66]. In this study, 604 patients were randomly allocated to either the 5-day regimen or the 1-day regimen group. The incidences of SSI in patients treated with the 5-day regimen and 1-day regimens were 15% and 17%, respectively (hazard ratio (HR) = 0.93, *p* = 0.73). The incidences of antibiotics-related adverse events in the 5-day regimen group and 1-day regimen group were 5% and 2%, respectively (HR = 3.24, *p* = 0.02). Although the randomized trial showed no significant benefits of a longer period of antibiotic administration, the duration of antibiotic use should be determined based on the risk of SSI in each case.

Bacteria generate biofilms on the surfaces of implants, and antibiotics cannot reach the bacteria through the biofilms. Most cases of infection around implants require the removal of the implants. Implant-related infections were significantly associated with amputation (OR = 24.0, *p* < 0.001) and worse functional outcomes (OR = 0.01, *p* < 0.001), and the success rate of infection treatment without the removal of the implant was only 4.5% [31]. Therefore, the prevention of infection on implant surfaces is required to reduce the risk of SSI in patients who undergo surgical treatment using implants. Recently, antibacterial coating techniques (such as silver coating, iodine coating, and antibiotics coating) have been developed (Table 3) [67,68,69,70], and some basic and clinical studies have proven the safety and efficacy of these various coating techniques [71]. These techniques are thought to reduce the social and economic burden of implant-related infections in orthopedic surgeries. 

The antibacterial activity of silver mostly depends on its ability to interfere with bacterial cell membrane permeability and cellular metabolism [71]. Silver coatings have been used on tumor prostheses [72,73]. Fiore et al. conducted a meta-analysis on SSI in patients treated with silver-coated megaprostheses [74]. In this study, the patients with the silver-coated megaprostheses had an infection rate of 9.2% after primary surgery, whereas the patients with uncoated megaprostheses had an infection rate of 11.2%. In contrast, the infection rate after revision megaprosthesis was 13.7% in patients with silver-coated megaprostheses and 29.2% in patients with uncoated megaprostheses (*p* = 0.019). This study suggests that the silver-coated megaprosthesis is effective in reducing the incidence of SSI in patients at a high risk of postoperative infection. In a retrospective study involving 51 patients with bone sarcoma who underwent megaprosthesis replacement, the infection rates in patients with silver-coated megaprostheses and control patients were found to be 6% and 18%, respectively (*p* = 0.062) [70]. In another retrospective case-control study, the clinical outcomes of silver-coated tumor prostheses in 85 patients were compared with 85 matched controls [67]. The infection rates in the silver-coating and control groups were 12% and 22%, respectively (*p* = 0.033). After debridement, antibiotics administration, and implant retention in patients with SSI, the success (infection control) rates in patients with silver-coated implants and control patients were 70% and 32%, respectively (*p* = 0.048). Parry et al. compared the clinical outcomes of silver-coated endoprostheses in 89 patients at high risk for SSI with the outcomes of non-silver-coated endoprostheses in 305 patients [75]. Although the silver-coated prosthesis group had a higher risk of SSI, no significant difference in infection rates was reported between the silver-coated endoprosthesis (12%) and uncoated endoprosthesis (8%) groups (*p* = 0.154). Although silver-coating seems to be effective in the prevention of SSI, it has some disadvantages, such as the cytotoxicity of silver ions to bone cells, the incomplete protection of the implant, and the high cost of the technique [76,77].

Povidone-iodine can be used as an electrolyte to form an adhesive porous anodic oxide (which retains the antibacterial properties of iodine) [78]. In basic research, an iodine coating showed a good antibiofilm effect, a long antibacterial effect, good osteoconductivity, and safety to human cells [78]. In clinical studies, iodine-coated titanium showed a preventive effect against SSI [68,69,78]. Tsuchiya et al. investigated the effect of iodine-coated implants in 222 patients with immune-compromised conditions or postoperative infection [69]. Iodine-coated implants were used to prevent infection in 158 patients and treat active infections in 64 patients. Among patients in whom iodine-coated implants were used to prevent infection, 1.9% developed acute infection. All patients with active infection could be treated successfully without implant removal. In their study, no cytotoxicity or adverse effects were observed. Shirai et al. investigated the effect of iodine-coated implants on the infection rates in patients who underwent tumor resection and reconstruction using tumor-bearing frozen autografts [79]. They reported that 10 of 62 (16%) patients treated with uncoated implants had deep infections, whereas only 1 of 38 (3%) patients treated with iodine-coated implants had a deep infection (*p* = 0.032). In another study investigating the predictive factors for SSI in patients who underwent malignant bone tumor resection and reconstruction, iodine-coated implants significantly reduced the risk of SSI (OR = 0.3) [26].

Although there are limited reports on the effectiveness of antibiotics-coated implants in patients with bone tumors, several reports exist on the effectiveness of these implants in the treatment of fractures [80,81]. Fuchs et al. reported that no implant-related infections were observed in 21 patients treated with gentamicin poly (D, L-lactide)-coated intramedullary nails for closed or open tibial fractures, except for one patient who had wound-healing difficulties [80]. Metsemakers et al. investigated clinical outcomes of gentamicin-coated intramedullary nails in 16 patients with open tibial fractures, including 11 acute fractures and 5 revision cases [81]. In their study, no patient had a deep infection after surgery using gentamicin-coated nails. However, four (25%) patients had nonunion. In a multicenter prospective study involving 99 patients with fresh tibial fractures or who underwent revision surgeries for nonunion, deep infection or osteomyelitis was observed in 7.2% of patients with fresh fractures and 8% of patients who underwent revision surgeries [82]. However, this technique has some disadvantages: its availability is limited only to patients with tibial defects, and its effect is reduced in the event of gentamicin resistance [71].

**Table 3 cancers-14-04527-t003:** Antibacterial coating techniques for the prevention of SSI during bone tumor surgery.

Study	N	Groups	Clinical Outcomes	Ref.
Silver-coated endoprosthesis	170	Silver: 85 Uncoated: 85	Infection rate: 11.8% in the silver-coated group and 22.4% in the control group (*p* = 0.033) Success rate of DAIR: 70% in the silver-coated group and 31.6% in the control group (*p* = 0.048)	[67]
Silver-coated endoprosthesis in patients with sarcoma	125	Silver: 51 Uncoated: 74	Infection rate: 5.9% in the silver-coated group and 17.6% in the titanium group (*p* = 0.062)	[70]
Silver-coated endoprosthesis in patients with sarcoma or GCTB	394	Silver: 89 Uncoated: 305	Infection rates: 12.4% in the silver-coated group and 7.5% in the non-silver group (*p* = 0.154)	[75]
Silver-coated endoprosthesis in two-stage revision for PJI	68	Silver: 29Uncoated: 39	Reinfection rate: 10.3% in the silver-coated group and 17.5% in the uncoated group (*p* = 0.104)	[83]
Silver-coated endoprosthesis in patients with proximal femur sarcoma	99	Silver: 64 patients Uncoated: 35	Infection rates: 9.4% in the silver-coated group and 14.3% in the uncoated group	[84]
Silver-coated endoprosthesis in patients with sarcoma or giant-cell tumor in the proximal tibia	98	Silver: 56 Uncoated: 42	Infection rates: 8.9% in the silver-coated group and 16.7% in the uncoated group (*p* = 0.247)	[85]
Silver-coated megaprosthesis in the proximal femur	68	Silver: 38 Uncoated: 30	Infection rates: 7.9% in the silver-coated group and 16.7% in the uncoated group	[86]
Iodine-coated implant in patients with malignant bone tumor	100	Iodine: 38 Uncoated: 62	Deep infection rates: 3% in iodine-coated implants and 16% in non-coated implants (*p* = 0.032)	[79]
Iodine-coated implant in patients with malignant bone tumor	302	Iodine: 66 Uncoated: 236	Iodine-coated implants reduced the risk of SSI (OR = 0.3, *p* = 0.039)	[26]

PJI, prosthetic joint infection; GCTB, giant cell tumor of bone; DAIR, debridement, antibiotics, and implant retention; OR, odds ratio.

## 5. Conclusions

According to the previous reports, tumors in the pelvis or tibia, leukocytopenia, a long operative time, and the use of implant/bone grafts are considered as risk factors for SSI in patients with bone tumors. Surgical planning based on a preoperative risk assessment of SSI, implants with antibacterial coatings, and efforts to reduce operative time are recommended to prevent SSI.

## Figures and Tables

**Table 1 cancers-14-04527-t001:** Standard definitions of SSI.

Criterion	CDC [38]	SISG [35]	NPS [34]	PHLS [40]
Purulent drainage	SI/DI/OS	✓		✓
Painful spreading erythema indicative of cellulitis		✓		
Isolation of bacteria from cultures	SI/OS	✓	✓	
Deliberate opening of the incision by the surgeon	SI/DI			
Spontaneous incisional wound dehiscence	DI			
Fever	DI	✓	✓	
Pain	SI/DI	✓	✓	
Tenderness	SI/DI	✓		
Swelling or redness	SI	✓		
Heat	SI			
Persisting elevation of the erythrocyte sedimentation rate		✓	✓	
Abscess or other evidence of infection	DI/OS			
Surgeon’s/physician’s diagnosis	SI/DI/OS			

CDC, the Center for Disease Control: SISG, the Surgical Infection Society Study Group: NPS, the 2nd UK National Prevalence Survey study group: SI, superficial incisional: DI, deep incisional: OS, organ/space.

**Table 2 cancers-14-04527-t002:** Risk factors for SSI in bone tumor surgery.

Patients	N	Risk Factors	Ref.
Bone tumors	844	Pelvic tumor (OR = 3.42, *p* = 0.044) and use of an implant (OR = 9.28, *p* = 0.006)	[43]
Malignant bone tumors	302	Pelvic tumor (OR = 3.4, *p* = 0.044) and operative time (OR = 3.4, *p* = 0.022)	[26]
Pelvic tumors	270	Pelvic reconstruction (OR = 0.51, *p* = 0.033)	[27]
Musculoskeletal tumors	1521	Age (OR = 1.18, *p* < 0.001), total number of preceding procedures (OR = 1.2, *p* < 0.01), preexisting implants (OR = 1.94, *p* = 0.001), infection at another site on the date of the surgery (OR = 4.13, *p* < 0.01), hip region affected (OR = 1.96, *p* < 0.001), and duration of the procedure (OR = 1.2, *p* < 0.01)	[41]
Malignant bone tumors of the extremities	256	Tibial tumor (OR = 6.04, *p* < 0.001) and operative time (OR = 3.3, *p* = 0.027)	[44]
Allograft reconstruction after proximal tibia resection	32	BMI (OR = 1.40, *p* < 0.05) and preoperative WBC (OR = 0.30, *p* < 0.05)	[51]
Reconstruction with massive allograft in long bones	673	Tibia allograft (OR = 3.17, *p* < 0.001), male patients (OR = 1.92, *p* < 0.029), procedures performed in a conventional operating room (OR = 3.15, *p* < 0.002), and the long-term use of postoperative antibiotics (OR = 2.25, *p* < 0.041)	[47]
Musculoskeletal tumors	110	Blood transfusion (*p* = 0.007) and obesity (*p* = 0.016)	[50]

OR, odds ratio; BMI, body mass index; WBC, white blood cell.

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
