# Peer review of "Surgical Site Infection after Bone Tumor Surgery: Risk Factors and New Preventive Techniques"

_cancers, 2022, doi:10.3390/cancers14184527_

Round 1
Reviewer 1 Report
In this review, the Authors attempted to discuss predictive factors of surgical site infections after the management of bone tumors and new prophylactic techniques.Although the topic might be very interesting, the paper is very disorganized and provide confusing information, thus not adding anything to the existing Literature.
This is a narrative review which analyze singular risk factors individually. For example, age might be a risk factor only in particular sites. Also, Radiotherapy effects on the pelvis can be extremely different than on other sites.
Introduction: Please try to provide more detailed data. I would add SSI incidence according to orthopaedic technique.
A table reporting definitions of SSI would help.
Tables are not very informative. Much information missing, in particular when dealing with silver and iodine implants.
Many updated references missing, in particular regarding silver coated implant.
Author Response
Comment 1. Introduction: Please try to provide more detailed data. I would add SSI incidence according to orthopaedic technique.
Response:
- Thank you for your valuable comments.
In Introduction section, detailed data were added as follows:
The incidence of SSI in patients who underwent orthopaedic surgeries were reported as 0.2–4.2%. The incidence of SSI in total hip arthroplasty was reported as 0.2%, whereas the incidences of SSI in spine surgeries were reported as 3.4–4.2%. On the other hand, high incidences of SSI have been reported in patients with malignant bone tumors. In patients who underwent megaprosthesis after malignant bone tumor resection, 9.3–20.4% of the patients had SSI. SSI rates after biological reconstruction were reported as 11.9–13.5%.
References
- Shapiro, L. M.; Graham, L. A.; Hawn, M. T.; Kamal, R. N., Quality Reporting Windows May Not Capture the Effects of Surgical Site Infections After Orthopaedic Surgery. J Bone Joint Surg Am 2022, 104 (14), 1281-1291.
- Matsumoto, H.; Larson, E. L.; Warren, S. I.; Hammoor, B. T.; Bonsignore-Opp, L.; Troy, M. J.; Barrett, K. K.; Striano, B. M.; Li, G.; Terry, M. B.; Roye, B. D.; Lenke, L. G.; Skaggs, D. L.; Glotzbecker, M. P.; Flynn, J. M.; Roye, D. P.; Vitale, M. G., A Clinical Risk Model for Surgical Site Infection Following Pediatric Spine Deformity Surgery. J Bone Joint Surg Am 2022, 104 (4), 364-375.
- Bourget-Murray, J.; Bansal, R.; Soroceanu, A.; Piroozfar, S.; Railton, P.; Johnston, K.; Johnson, A.; Powell, J., Assessment of risk factors for early-onset deep surgical site infection following primary total hip arthroplasty for osteoarthritis. J Bone Jt Infect 2021, 6 (9), 443-450.
- Staals, E. L.; Sambri, A.; Campanacci, D. A.; Muratori, F.; Leithner, A.; Gilg, M. M.; Gortzak, Y.; Van De Sande, M.; Dierselhuis, E.; Mascard, E.; Windhager, R.; Funovics, P.; Schinhan, M.; Vyrva, O.; Sys, G.; Bolshakov, N.; Aston, W.; Gikas, P.; Schubert, T.; Jeys, L.; Abudu, A.; Manfrini, M.; Donati, D. M., Expandable distal femur megaprosthesis: A European Musculoskeletal Oncology Society study on 299 cases. J Surg Oncol 2020.
- Pala, E.; Trovarelli, G.; Calabro, T.; Angelini, A.; Abati, C. N.; Ruggieri, P., Survival of modern knee tumor megaprostheses: failures, functional results, and a comparative statistical analysis. Clin Orthop Relat Res 2015, 473 (3), 891-9.
- Lee, S. Y.; Jeon, D. G.; Cho, W. H.; Song, W. S.; Kim, B. S., Are Pasteurized Autografts Durable for Reconstructions After Bone Tumor Resections? Clin Orthop Relat Res 2018, 476 (9), 1728-1737.
- Araki, Y.; Yamamoto, N.; Hayashi, K.; Takeuchi, A.; Miwa, S.; Igarashi, K.; Higuchi, T.; Abe, K.; Taniguchi, Y.; Yonezawa, H.; Morinaga, S.; Asano, Y.; Tsuchiya, H., Clinical outcomes of frozen autograft reconstruction for the treatment of primary bone sarcoma in adolescents and young adults. Sci Rep 2021, 11 (1), 17291.
Comment 2. A table reporting definitions of SSI would help.
Response:
The following table reporting definitions of SSI was added to the manuscript.
Table 1. Standard definitions of SSI
|
Criterion |
CDC |
SISG |
NPS |
PHLS |
|
Purulent drainage |
SI/DI/OS |
✓ |
|
✓ |
|
Painful spreading erythema indicative of cellulitis |
|
✓ |
|
|
|
Isolation of bacteria from cultures |
SI/OS |
✓ |
✓ |
|
|
Deliberate opening of the incision by the surgeon |
SI/DI |
|
|
|
|
Spontaneous incisional wound dehiscence |
DI |
|
|
|
|
Fever |
DI |
✓ |
✓ |
|
|
Pain |
SI/DI |
✓ |
✓ |
|
|
Tenderness |
SI/DI |
✓ |
|
|
|
Swelling or redness |
SI |
✓ |
|
|
|
Heat |
SI |
|
|
|
|
Persisting elevation of the erythrocyte sedimentation rate |
|
✓ |
✓ |
|
|
Abscess or other evidence of infection |
DI/OS |
|
|
|
|
Surgeon’s/physician’s diagnosis |
SI/DI/OS |
|
|
|
CDC, the Center for Disease Control; SISG, the Surgical Infection Society study Group; NPS, the 2nd UK National Prevalence Survey study group; PHLS, the Public Health Laboratory Service; SI, superficial incisional: DI, deep incisional: OS, organ/space
Comment 3:
Tables are not very informative. Much information missing, in particular when dealing with silver and iodine implants.
Many updated references missing, in particular regarding silver coated implant.
Response:
Thank you for your pointing this out. Recent studies on silver-coating were added to the Table.
|
Study |
N |
Groups |
Clinical outcomes |
Ref. |
|
Silver-coated endoprosthesis |
170 |
Silver: 85 Non-coating: 85 |
Infection rate: 11.8% in the silver-coated group and 22.4% in the control group (P = 0.033) Success rate of DAIR: 70% in the silver-coated group and 31.6% in the control group (P = 0.048) |
66 |
|
Silver-coated endoprosthesis in patients with sarcoma |
125 |
Silver: 51 Non-coating: 74 |
Infection rate: 17.6% in the titanium group and 5.9% in the silver-coated group (P = 0.062) |
69 |
|
Silver-coated endoprosthesis in patients with sarcoma or GCTB |
394 |
Silver: 89 Non-coating: 305 |
Infection rates: 12.4% in the silver-coated group and 7.5% in the non-silver group (P = 0.154) |
73 |
|
Silver-coated endoprosthesis in two-stage revision for PJI |
68 |
Silver: 29 |
Reinfection rate: 10.3% in the silver-coated group and 17.5% in uncoated group (P = 0.104) |
81 |
|
Silver-coated endoprosthesis in patients with proximal femur sarcoma |
99 |
Silver: 64 patients Non-coating: 35 |
Infection rates: 14.3% in uncoated group and 9.4% in the silver-coated group |
82 |
|
Silver-coated endoprosthesis in patients with sarcoma or giant-cell tumor in the proximal tibia |
98 |
Silver: 56 Non-coating: 42 |
Infection rates: 8.9% in the silver-coated group and 16.7% in uncoated group (P = 0.247) |
83 |
|
Silver-coated megaprosthesis in the proximal femur |
68 |
Silver: 38 Non-coating: 30 |
Infection rates: 7.9% in the silver-coated group and 16.7% in uncoated group |
84 |
|
Iodine-coated implant in patients with malignant bone tumor |
100 |
Iodine: 38 Non-coating: 62 |
Deep infection rates: 16% in non-coated implants and 3% in iodine-coated implants (P = 0.032) |
77 |
|
Iodine-coated implant in patients with malignant bone tumor |
302 |
Iodine: 66 Non-coating: 236 |
Iodine-coated implants reduced the risk of SSI (OR = 0.3, P = 0.039) |
26 |
PJI, prosthetic joint infection: GCTB, giant cell tumor of bone
References:
- Sambri, A.; Zucchini, R.; Giannini, C.; Zamparini, E.; Viale, P.; Donati, D. M.; De Paolis, M., Silver-coated (PorAg((R))) endoprosthesis can be protective against reinfection in the treatment of tumor prostheses infection. Eur J Orthop Surg Traumatol 2020, 30 (8), 1345-1353.
- Streitbuerger, A.; Henrichs, M. P.; Hauschild, G.; Nottrott, M.; Guder, W.; Hardes, J., Silver-coated megaprostheses in the proximal femur in patients with sarcoma. Eur J Orthop Surg Traumatol 2019, 29 (1), 79-85.
- Hardes, J.; Henrichs, M. P.; Hauschild, G.; Nottrott, M.; Guder, W.; Streitbuerger, A., Silver-Coated Megaprosthesis of the Proximal Tibia in Patients With Sarcoma. J Arthroplasty 2017, 32 (7), 2208-2213.
- Donati, F.; Di Giacomo, G.; D'Adamio, S.; Ziranu, A.; Careri, S.; Rosa, M.; Maccauro, G., Silver-Coated Hip Megaprosthesis in Oncological Limb Savage Surgery. Biomed Res Int 2016, 2016, 9079041.
Reviewer 2 Report
This is a very intersting paper about risk factors of SSI after bone tumor surgery.
I don't have any concern or suggestion for the paper presented.
Author Response
Thank you for your comments. The manuscript was corrected according to the other reviewer’s comments.
Reviewer 3 Report
The present article is a comprehensive review article on surgical site infections after surgery for bone tumors. The manuscript is well-structured and very informative.
Minor considerations:
1. page 2-line 67-68: "Bruce et al reported that there are 41 definition of SSI.." I would prefer more vaguely "nurmerous/several definitions" instead of the exact number (41 definitions) because not every one of the 41 definitions might be formal/typical and also the number changes.
2. page 2: "Pelvic tumors are thought to be associated with an increased risk of SSI because of the reported high incidence of SSI in this location (10–37%)"
please rephrase (too wordy)
3. In the section "Current practices in the prophylaxis of SSI and emerging techniques" please briefly discuss if there are international recommendations/guidelines on prophylactic perioperative antibiotic use.
Author Response
Comment 1. page 2-line 67-68: "Bruce et al reported that there are 41 definition of SSI.." I would prefer more vaguely "nurmerous/several definitions" instead of the exact number (41 definitions) because not every one of the 41 definitions might be formal/typical and also the number changes.
Response:
Thank you for recommendation. The sentence was corrected as “Bruce et al reported that there are numerous definitions of SSI…”.
Comment 2. page 2: "Pelvic tumors are thought to be associated with an increased risk of SSI because of the reported high incidence of SSI in this location (10–37%)"
please rephrase (too wordy)
Response:
Thank you for pointing this out. The sentence was corrected as follows:
Because high incidence of SSI (10–37%) has been reported in pelvic tumors, this location is thought to be associated with an increased risk of SSI.
Comment 3. In the section "Current practices in the prophylaxis of SSI and emerging techniques" please briefly discuss if there are international recommendations/guidelines on prophylactic perioperative antibiotic use.
Response:
Thank you for your recommendation. The following sentences were added to the section “Current practices in the prophylaxis of SSI and emerging techniques”.
In the International Consensus Meeting on Musculoskeletal Infection, there is no recommendation to adjust type, dose, and duration of antibiotic prophylaxis in patients undergoing oncologic endoprosthetic reconstruction from that which is routinely administered in conventional total joint arthroplasty.
Reference:
Parvizi, J.; Gehrke, T., Proceedings of the Second International Consensus Meeting on Musculoskeletal Infection. Data Trace Publishing Company: 2018.
Round 2
Reviewer 1 Report
The Authors made good efforts in the attempt to ameliorate their paper.
Still missing a recent review on silver implants (Fiore et al Eur J Orthop Surg Traumatol. 2021 doi: 10.1007/s00590-020-02779-z.)
Author Response
Thank you for your suggestion. The following sentences were added to the section “Current practices in the prophylaxis of SSI and emerging techniques”.
Fiore et al. conducted a meta-analysis on SSI in patients treated with silver-coated megaprostheses. In this study, the patients with the silver-coated megaprosthesis had an infection rate of 9.2% after primary surgery, whereas the patients with uncoated megaprosthesis had an infection rate of 11.2%. In contrast, the infection rate after revision megaprosthesis was 13.7% in patients with silver-coated megaprosthesis and 29.2% in patients with uncoated megaprosthesis (P = 0.019). This study suggests that the silver-coated megaprosthesis is effective in reducing the incidence of SSI in patients at a high risk of postoperative infection.